# Sex-dependent gastrointestinal colonization resistance to MRSA is microbiota and Th17 dependent

**Alannah Lejeune**[1,2], **Chunyi Zhou**[1,2], **Defne Ercelen**[3], **Gregory Putzel**[1,4], **Xiaomin Yao**[2], **Alyson R Guy**[5], **Miranda Pawline**[3], **Magdalena Podkowik**[2,4], **Alejandro Pironti**[1,4], **Victor J Torres**[1,6]*, **Bo Shopsin**[1,2,4]*, **Ken Cadwell**[7,8]*

[1]Department of Microbiology, New York University School of Medicine, New York, United States; [2]Department of Medicine, Division of Infectious Diseases, New York University School of Medicine, New York, United States; [3]Department of Medicine, Division of Gastroenterology and Hepatology, New York University Langone Health, New York, United States; [4]Antimicrobial-Resistant Pathogens Program, New York University School of Medicine, New York, United States; [5]NYU-Regeneron Veterinary Postdoctoral Training Program in Laboratory Animal Medicine, Division of Comparative Medicine, New York University School of Medicine, New York, United States; [6]Department of Host-Microbe Interactions, St. Jude Children's Research Hospital, Memphis, United States; [7]Department of Medicine, Division of Gastroenterology and Hepatology, University of Pennsylvania Perelman School of Medicine, Philadelphia, United States; [8]Department of Pathobiology, University of Pennsylvania Perelman School of Veterinary Medicine, Philadelphia, United States

*For correspondence:
victor.torres@stjude.org (VJT);
Bo.Shopsin@nyulangone.org
(BS);
Ken.Cadwell@Pennmedicine.
upenn.edu (KC)

## eLife Assessment

This **fundamental** study highlights potential mechanisms underlying the sex-dependent bias in susceptibility to gut colonization by Methicillin-resistant *Staphylococcus aureus* (MRSA). The evidence supporting the conclusion is **compelling**. The work will interest biologists who study intestinal infection and immunity.

**Abstract** Gastrointestinal (GI) colonization by methicillin-resistant *Staphylococcus aureus* (MRSA) is associated with a high risk of transmission and invasive disease in vulnerable populations. The immune and microbial factors that permit GI colonization remain unknown. Male sex is correlated with enhanced *Staphylococcus aureus* nasal carriage, skin and soft tissue infections, and bacterial sepsis. Here, we established a mouse model of sexual dimorphism during GI colonization by MRSA. Our results show that in contrast to male mice that were susceptible to persistent colonization, female mice rapidly cleared MRSA from the GI tract following oral inoculation in a manner dependent on the gut microbiota. This colonization resistance displayed by female mice was mediated by an increase in IL-17A+ CD4+ T cells (Th17) and dependent on neutrophils. Ovariectomy of female mice increased MRSA burden, but gonadal female mice that have the Y chromosome retained enhanced Th17 responses and colonization resistance. Our study reveals a novel intersection between sex and gut microbiota underlying colonization resistance against a major widespread pathogen.

## Introduction

Methicillin-resistant *Staphylococcus aureus* (MRSA) is a major global health concern due to its multi-drug resistance, wide range of infections, and high morbidity and mortality rates (*David and Daum, 2010*; *Seybold et al., 2006*; *Boucher and Corey, 2008*; *de Kraker et al., 2011*). Gastrointestinal (GI) *S. aureus* colonization is present in an estimated 20% of the healthy population (*Acton et al., 2009*; *Gagnaire et al., 2017*), and an estimated 1–6% are persistently colonized with MRSA (*Acton et al., 2009*; *Gagnaire et al., 2017*). MRSA carriage increases the likelihood of invasive infections, especially while in the hospital or post discharge (*Huang, 2019*; *von Eiff et al., 2001*). Intestinal carriage is associated with higher rates of infections and bacteremia than nasal carriage alone (*de Kraker et al., 2011*; *Squier et al., 2002*). The risk of transmission to other individuals in the hospital and community settings through fomite spread is increased by intestinal carriage (*Boyce et al., 2007*; *Bhalla et al., 2007*).

A better understanding of why a subset of individuals are susceptible to colonization may inform decolonization strategies (*Acton et al., 2009*; *Gagnaire et al., 2017*; *Huang, 2019*). Although little is known mechanistically, population-level studies have identified correlates of colonization. For example, male sex is correlated with *S. aureus* nasal carriage, skin and soft tissue infections, and bacterial sepsis in adulthood (*Nowak et al., 2017*; *Humphreys et al., 2015*; *Castleman et al., 2018*; *Tacconelli and Foschi, 2017*). High free testosterone levels are a risk factor for *S. aureus* throat carriage in women (*Nowak et al., 2017*). However, experimental systems are necessary to clarify to what extent male sex as a risk factor for carriage is biological or behaviorally based.

Sex steroid hormones and sex chromosomes can modulate the scale and type of immune response (*Schurz et al., 2019*; *Klein and Flanagan, 2016*; *Jaillon et al., 2019*). Sex hormones such as estrogen and testosterone can directly influence immune cell activation, proliferation, and cytokine response through receptor signaling (*Vázquez-Martínez et al., 2018*; *Dias et al., 2022*; *Fuseini et al., 2019*; *Chi et al., 2024*; *Li et al., 2024*). In general, males are more susceptible to GI and respiratory infections, whereas females are more affected by autoimmune diseases in part due to the pro-inflammatory effect of estradiol, but these responses are tissue and cell specific (*Klein and Flanagan, 2016*; *Vázquez-Martínez et al., 2018*). In the few studies examining GI MRSA colonization in mice, the focus has been on the adaptation of *S. aureus* to the host (*Misawa et al., 2015*; *Kernbauer et al., 2015*; *Piewngam et al., 2018*; *Flaxman et al., 2017*). These studies generally rely on depletion of the mouse microbiota with antibiotics to establish long-term colonization (*Misawa et al., 2015*; *Kernbauer et al., 2015*; *Gries et al., 2005*). To investigate mechanisms of colonization resistance in a setting with an intact microbiota, we established a GI colonization model that does not rely on antibiotic treatment, better recapitulating MRSA colonization in a healthy host. We report that following MRSA oral inoculation, female mice with a nonpermissive microbiota were resistant to sustained colonization compared with male mice that remain persistently colonized. Colonization resistance displayed by female mice was mediated by an increase in T helper 17 (Th17) cells and associated with sex hormones rather than sex chromosomes. Thus, our study demonstrates a role for the Th17 response and microbiota in GI colonization resistance against MRSA, while also highlighting the sex-dependent susceptibility to this major pathogen.

## Results

### Female mice are protected from MRSA GI colonization in a microbiota-dependent manner

We and others have shown that inbred laboratory mice from different sources, or even the same institution, can display substantial differences in mucosal immune responses and susceptibility to GI colonization by microorganisms (*Jang et al., 2023*; *Ramanan et al., 2014*; *Cadwell et al., 2010*; *Moon et al., 2015*; *Ivanov et al., 2009*; *Mamantopoulos et al., 2017*). Therefore, when we set out to establish a model of GI MRSA colonization, we examined two sources of C57BL/6J (B6) mice - those bred in Jackson Laboratory (JAX) and genetically identical mice bred within our institutional animal facility (referred to as NYU mice). Adult B6 mice received a single oral gavage with $10^8$ colony forming units (CFU) of MRSA USA300 LAC, the predominant MRSA clone responsible for community-associated MRSA infections and a growing number of hospital-acquired infections in the United States (*Seybold et al., 2006*). Prior to inoculation, we confirmed the absence of pre-existing *S. aureus*

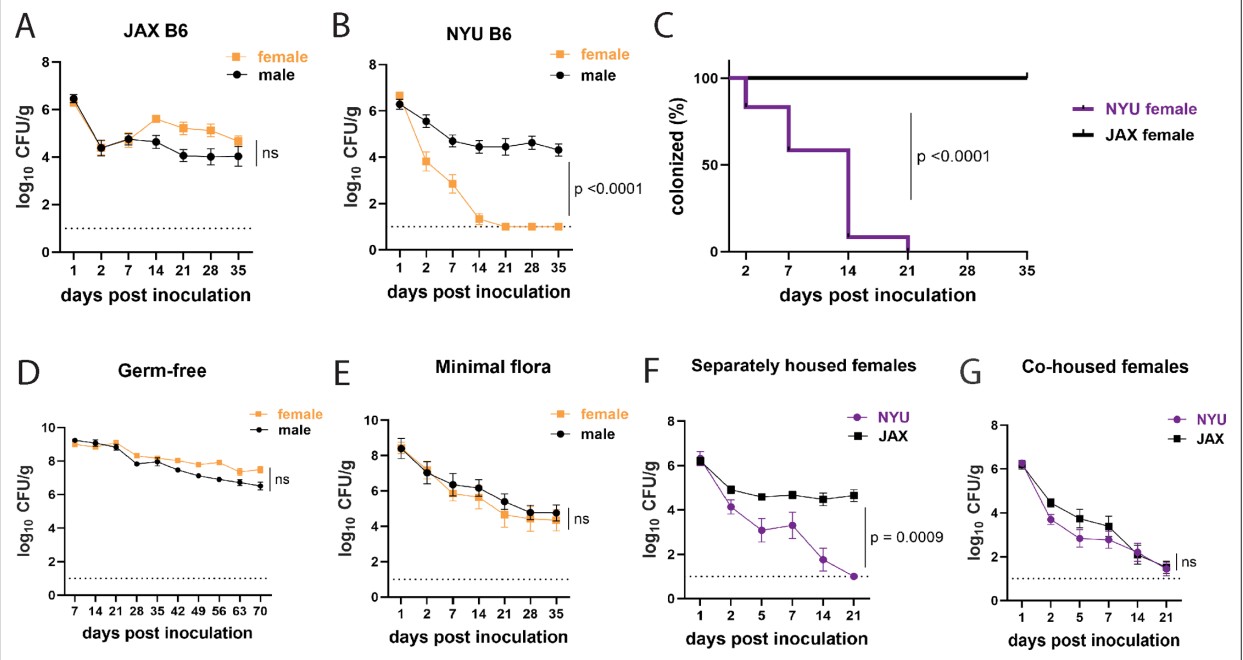

**Figure 1.** Female mice are protected from methicillin-resistant *S. aureus* (MRSA) gastrointestinal colonization in a microbiota-dependent manner. (**A**) MRSA colony forming units (CFU) per gram of stool following oral inoculation of B6 mice purchased from Jackson Laboratory (JAX). Males n=10, females n=13. (**B**) MRSA CFU in stool following oral inoculation of B6 mice generated from breeders in the NYU animal facility. Males n=13, females n=13. (**C**) Proportion of JAX and NYU female mice with detectable MRSA gastrointestinal (GI) colonization over time. (**D**) MRSA CFU stool burden following oral inoculation of germ-free mice. Males n=8, females n=7. (**E**) MRSA CFU in stool following oral inoculation of mice with a defined minimal microbiota. Males n=12, females n=11. (**F**) MRSA CFU stool burden of female NYU or JAX mice housed separately or (**G**) co-housed with each other. NYU n=9, JAX n=11, co-housed NYU n=14, co-housed JAX n=14. Data points represent mean ± SEM from at least two independent experiments. Statistical analysis: area under the curve followed by a two-tailed t-test for (**A**), (**B**), (**D–G**) and Log-rank Mantel-Cox test for (**C**). ns: not significant.

The online version of this article includes the following figure supplement(s) for figure 1:

**Figure supplement 1.** Mice inoculated with methicillin-resistant *S. aureus* (MRSA) do not display signs of disease.

colonization by plating stool from individual mice on ChromAgar plates which select for *S. aureus* growth. We confirmed that mice inoculated with MRSA do not display signs of disease such as weight loss (*Figure 1—figure supplement 1A*). Consistent with previous findings (*Zhou et al., 2024*), we did not observe extraintestinal dissemination of bacteria or histological signs of intestinal inflammation (*Figure 1—figure supplement 1B–D*).

In JAX mice, we detected $10^4$–$10^5$ CFU per gram of stool of MRSA on day 2 that remained stable for at least 35 days post inoculation (dpi) (*Figure 1A*). However, we noticed bimodal distribution of the data for NYU mice where some mice remained stably colonized while detectable MRSA burden diminished rapidly in others. Segregation of the groups by sex revealed that NYU mice that displayed resistance to MRSA colonization were females. Despite originating from the same litters, male NYU mice displayed sustained colonization at levels similar to male and female JAX mice, while MRSA levels were already lower by day 2 and undetectable by 2–3 weeks post inoculation in female NYU mice (*Figure 1B and C*).

Commensal microbes that are constituents of the microbiota can inhibit *S. aureus* nasal, skin, and intestinal colonization (*Piewngam et al., 2018*; *Krismer et al., 2017*; *Nakatsuji et al., 2017*). In support of a role for commensal microbes in promoting colonization resistance in NYU female mice, we observed stable MRSA GI colonization in germ-free (GF) mice of both sexes (*Figure 1D*). We next examined GF mice that were colonized with a defined consortium of 15 bacterial strains representative of a mouse gut microbiota (Oligo-MM$_{12}$+FA3), previously shown to be sufficient to confer colonization resistance against the enteric pathogen *Salmonella enterica* serovar Typhimurium (*Brugiroux et al., 2016*). We observed sustained MRSA GI colonization and no sex difference in these minimal flora mice, indicating that additional intestinal commensals contribute to sex-dependent colonization resistance (*Figure 1E*). Given these results, we examined whether we could transfer the MRSA resistance

displayed by NYU female mice to permissive JAX female mice through co-housing mice in the same cage, which allows for the transfer of microbiota between mice (*Robertson et al., 2019*). JAX and NYU female mice were co-housed together for a week prior to MRSA inoculation and kept together for the duration of the experiment. When housed together, JAX female mice cleared MRSA colonization similarly to their NYU cage mates (*Figure 1F and G*), indicating that the microbiota of NYU female mice promotes resistance to colonization and that this protective property can be transferred.

## Microbiota is not sufficient to explain sex bias in MRSA GI colonization

To compare the microbiome composition of male and female NYU and JAX mice, we performed 16S ribosomal DNA sequencing of stool collected prior to inoculation with MRSA. Principal coordinates analysis (PCoA) of operational taxonomic units (OTUs) showed that samples were distinguished based on the source of mice (NYU vs JAX) rather than sex (*Figure 2A*). Overall alpha diversity as measured by Shannon index was increased in JAX mice compared to NYU mice prior to MRSA inoculation, as well as in JAX females compared to NYU females (*Figure 2B*). Alpha diversity by Shannon index between NYU males and females shows a similar spread when comparing sexes (*Figure 2B*). JAX females had higher abundance of Clostridiaceae and Lachnospiraceae family members, while NYU female microbiomes were dominated by the Muribaculaceae family (*Figure 2C*). The NYU male and female microbiomes clustered together (*Figure 2A*) and had similar relative taxonomic abundances (*Figure 2D*). These findings suggest that the observed sex bias in colonization resistance may not be due to microbiome composition alone.

As male and female mice are housed separately post weaning, and weaning is associated with microbiome changes (*Schloss et al., 2012*), we tested if the colonization resistance displayed by female NYU mice could be transferred to male NYU mice. Instead of co-housing male and female mice, which could introduce variables through social stress or changes in hormone levels due to mating (*Palanza et al., 2001*), we performed fecal microbiota transplantations (FMT) by repetitively gavaging recipient mice with donor stool. JAX female mice receiving an FMT from NYU female, but not male donor mice, displayed colonization resistance to MRSA. However, an FMT using NYU female donor stool was unable to confer colonization resistance to JAX male recipients (*Figure 2E*). Therefore, the microbiota contributes to the difference between female JAX and NYU mice, but other factors mediate the divergence of female and male NYU mice. From here on, we use NYU bred mice to investigate host factors that mediate this sex-specific colonization resistance.

## MRSA GI colonization elicits sex-dependent gene expression changes in the gut

To gain insight into the host response, we performed bulk RNA-seq analysis of cecal-colonic tissue obtained 2 dpi of NYU mice (both sexes) with either MRSA or PBS (mock), as 2 dpi is the time point at which MRSA burden begins to diverge between males and females. The lamina propria compartment, enriched with immune cells, and the epithelial cell fraction were isolated and sequenced separately. In the lamina propria, 795 genes were upregulated in both male and female mice following MRSA inoculation, with an additional 352 genes upregulated in females only and 648 genes upregulated in males only, as determined by a log2 fold change cutoff of 1.2 and a false discovery rate of 0.05 (*Figure 3—figure supplement 1A*). Following MRSA inoculation in both sexes, genes with the largest downregulation included those involved with immunoglobulins (*Ighv1-19*, *Igkv4-91*, *Bcam*), cytochrome p450 genes (*Cyp1a1*, *Cyp1b1*), and von Willebrand factor (*Vwf*), a secreted protein that MRSA binds to disrupt platelet recruitment and coagulation (*Steinert et al., 2020*; *Figure 3A*). Interestingly, constitutive cytochrome P4501A1 (*Cyp1a1*) expression impairs the intestinal immune response against enteric pathogens (*Schiering et al., 2017*), and its downregulation in mice inoculated with MRSA may be supportive of a mucosal immune response. Genes that were upregulated in the lamina propria following MRSA inoculation included those involved in T cell and neutrophil activity such as *Ccl28* (*Mohan et al., 2017*), *Sectm1b* (*Huyton et al., 2011*), *S100g* (*Donato, 2007*), *Lrrc19* (*Cao et al., 2016*), and *H2-q1* (*Anderson and Brossay, 2016*; *Figure 3A*).

Pathway analysis of the lamina propria fraction indicated that both males and females have downregulation of immune-related Th1, Th2, and granzyme A signaling pathways, and upregulation of oxidative phosphorylation, neutrophil extracellular trap formation, glucose metabolism, and xenobiotic metabolism pathways (*Figure 3B*). Downregulation of interleukin (IL)-4 and IL-13 pathway and

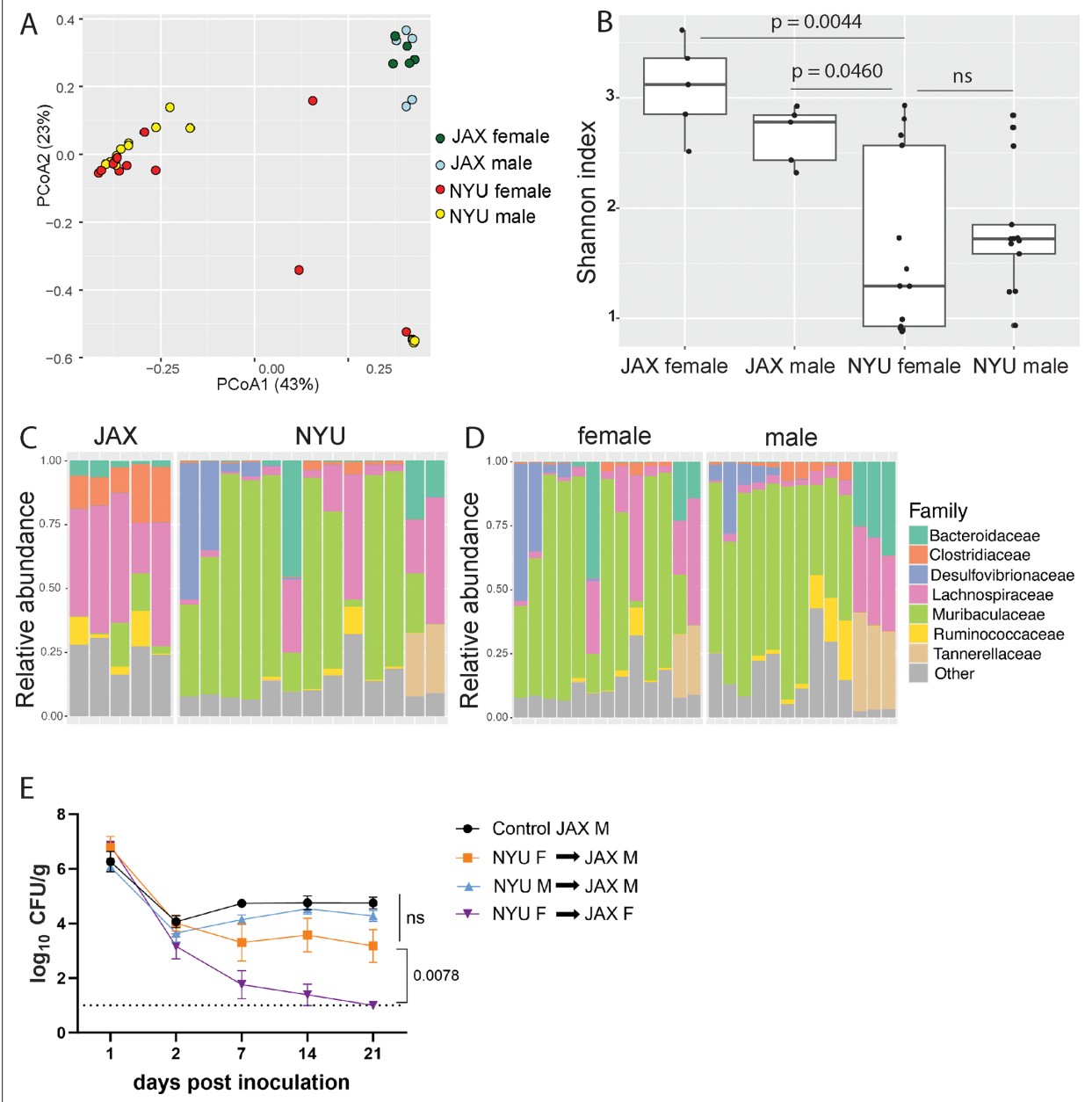

**Figure 2.** Microbiome is not sufficient to explain sex bias in methicillin-resistant *S. aureus* (MRSA) gastrointestinal (GI) colonization. (**A**) Principal coordinates analysis (PCoA) based on Bray-Curtis distances of 16S sequences obtained from stool of Jackson Laboratory (JAX) and NYU male and female mice prior to MRSA inoculation. Proportion of variance explained by each axes shown in parentheses. JAX M n=5, JAX F n=5, NYU M n=11, NYU F n=13. (**B**) Alpha diversity of the microbiomes of male and female JAX and NYU mice. Wilcoxon rank sum test used for pairwise statistical comparisons to NYU females; ns: not significant. (**C**) Relative abundance of bacterial families in female NYU and JAX mice prior to MRSA inoculation. (**D**) Relative abundance of bacterial families in male and female NYU mice prior to MRSA inoculation. (**E**) MRSA colony forming units (CFU) in stool following oral inoculation of JAX female (JAX F) and male (JAX M) recipients of fecal microbiota transplantations (FMT) from female (NYU F) and male (NYU M) donor mice compared with JAX male controls (control JAX M) that did not receive an FMT. Mean MRSA burden ± SEM. Area under the curve analysis+one-way ANOVA Sidak's multiple comparisons test for (**E**). Control M n=6, M+NYU F stool n=7, M+NYU M stool n=6, F+NYU F stool n=6. ns: not significant.

upregulation of retinoid X receptor and the pregnane X receptor pathways were specific to female mice. Unique to male mice was the downregulation of IL-5, IL-3, and GM-CSF signaling, and upregulation of the pyroptosis and ion channel transport pathways.

Known X and Y chromosome linked genes *Xist*, *Ddx3y*, *Eif2s3y*, *Kdm5d*, and *Uty* were differentially present in females and males as expected but were removed from male vs female analysis plots to

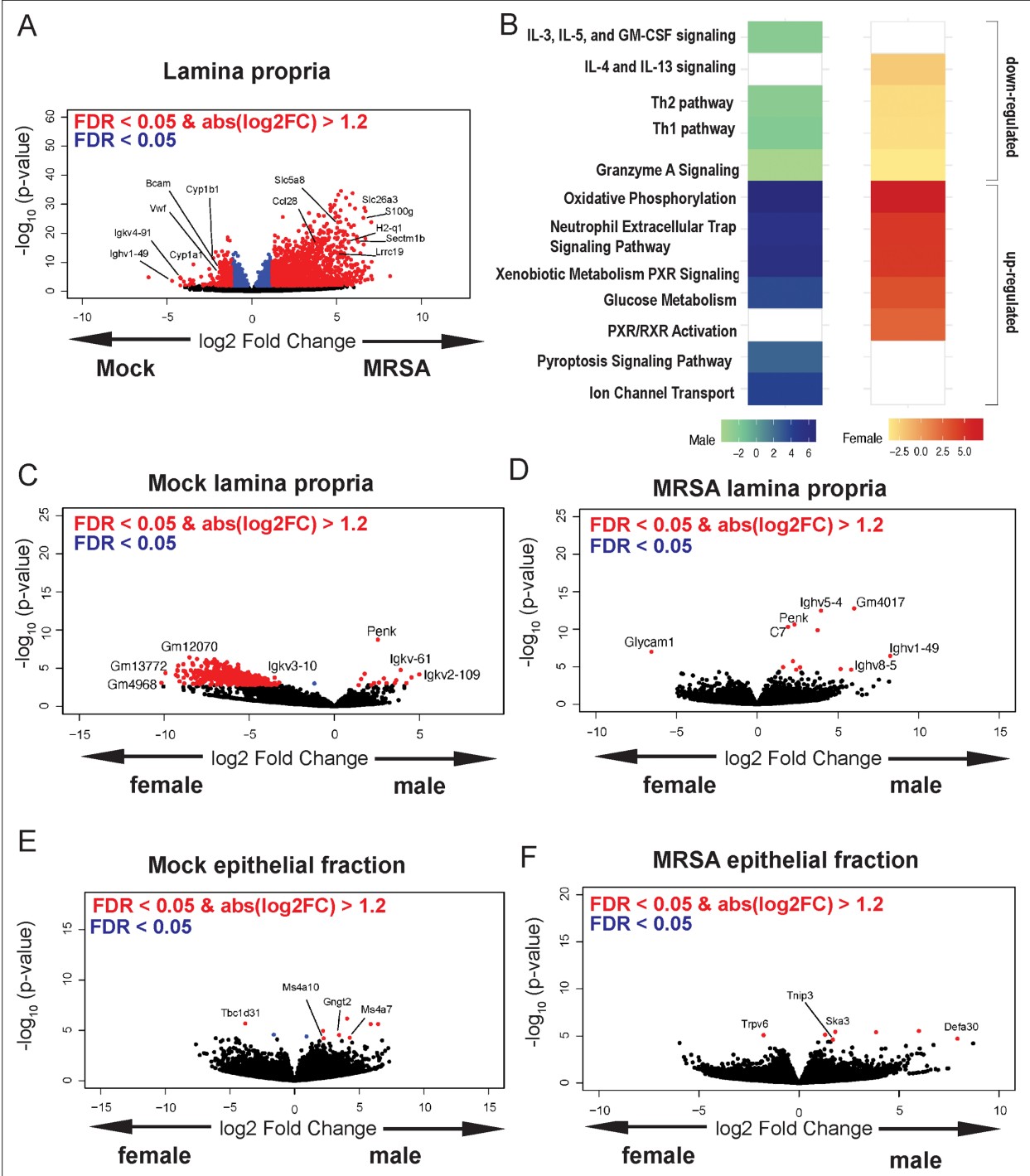

**Figure 3.** Methicillin-resistant *S. aureus* (MRSA) gastrointestinal (GI) colonization elicits sex-dependent gene expression changes in the gut. (**A**) Volcano plot of differentially expressed genes identified by RNA-seq analyses of the intestinal lamina propria of NYU mice 2 days post inoculation (dpi) with MRSA compared with phosphate-buffered saline (PBS) mock inoculated NYU mice. (**B**) Ingenuity Pathway Analysis (IPA) of downregulated and upregulated genes in the intestinal lamina propria upon MRSA inoculation of male and female mice. (**C**) Volcano plot of differentially expressed genes in the intestinal lamina propria comparing mock-treated males and females. (**D**) Volcano plot of differentially expressed genes in the intestinal lamina propria comparing males and females 2 dpi with MRSA. (**E**) Volcano plot of differentially expressed genes in the intestinal epithelial fraction of males and females prior to MRSA inoculation. (**F**) Volcano plot of differentially expressed genes in the intestinal epithelial fraction of males and females 2 dpi with MRSA. Four mice were used for each sequencing experimental condition. Genes shown in red have a false discovery rate (FDR) of <0.05 and an absolute log2 fold change (abslog2FC) of >1.2. Genes shown in blue have a FDR of <0.05. Genes linked to X and Y chromosomes were removed from volcano plots.

*Figure 3 continued on next page*

*Figure 3 continued*

The online version of this article includes the following figure supplement(s) for figure 3:

**Figure supplement 1.** Transcriptional analyses of the intestine following methicillin-resistant *S. aureus* (MRSA) inoculation.

highlight other differentially expressed genes. Other than known sex chromosome linked genes, when comparing male and female mice prior to MRSA inoculation, we observed several genes without a descriptive name or suggested pseudogenes such as *Gm12070*, *Gm13772*, and *Gm4968* (**Figure 3C**). In the MRSA condition, there were 13 genes differentially expressed in the intestinal lamina propria between male and female mice. *Glycosylation-dependent cell adhesion molecule-1* (*Glycam1*), which mediates lymphocyte trafficking to lymphoid tissues (**Brustein et al., 1992**), was upregulated in female mice (**Figure 3D**). Genes selectively upregulated in male mice 2 dpi included *Complement protein C7* and several immunoglobulin heavy chain variants (*Ighv5-4*, *Ighv1-49*, *Ighv8-5*).

In the intestinal epithelial fraction, we observed only 15 genes upregulated in males 2 dpi with MRSA, while 2037 were upregulated in females with no overlap between sexes (**Figure 3—figure supplement 1B**). Among the most differentially downregulated genes following MRSA inoculation when combining both sexes were *Irf5*, a pivotal transcription factor in the type I IFN pathway, *Csf2rb*, part of the receptor for IL-3, IL-5, and CSF signaling, and *Ciita*, a mediator of major histocompatibility complex (MHC) activation. Among those upregulated were genes involved in limiting inflammation like *Cav1*, involved in epithelial barrier maintenance, *Cfh*, a regulator of the complement pathway, and *Slpi* which functions to protect tissue from the detrimental consequences of inflammation (**Figure 3—figure supplement 1C**). Pathway analysis indicated that both sexes had upregulation of cell cycle and rRNA processing pathways and downregulation of neutrophil degranulation. Unique to females was upregulation of PTEN signaling and SUMOlyation of DNA damage response pathways (**Figure 3—figure supplement 1D**). There were few differentially expressed genes between males and females in the mock epithelial condition (**Figure 3E**) but following MRSA inoculation males had increased expression of the antimicrobial peptide *Defa30* and *Tnip3*, a negative regulator of NFκβ and Toll-like receptors (**Figure 3F**). Thus, MRSA inoculation elicits distinct transcriptional responses in males and females in both the lamina propria and epithelial fractions, and many of the differentially regulated genes have known functions in neutrophils and lymphocyte migration and function.

## CD4+ T cells mediate colonization resistance in female mice

Female mice have been shown to be protected from dermonecrosis in a model of invasive MRSA skin infection due to reduced expression of genes associated with the NLRP3 inflammasome (*Nlrp3* and *Il1β*) compared to males (**Castleman et al., 2018**). However, we did not observe sex-specific differences in NLRP3-associated transcripts with RNA-seq (**Figure 4—figure supplement 1A and B**) and female *Nlrp3*$^{-/-}$ mice displayed similar resistance to MRSA GI colonization as female *Nlrp3*$^{+/-}$ controls (**Figure 4—figure supplement 1C**). Thus, the mechanism of sex-specific resistance to MRSA colonization of the GI tract may be different from that observed during a skin infection where barrier breach triggers an exaggerated innate immune response that may cause more harm than benefit.

The T cell signature in the RNA-seq analyses of the intestinal lamina propria of MRSA colonized mice was unexpected given that the day 2 time point is earlier than what is typically required for an adaptive immune response. To test the role of lymphocyte responses in GI colonization resistance, we measured bacteria in stool following oral inoculation with MRSA of *recombination activating gene 2* (*Rag2*) deficient mice, which lack mature B and T cells. We observed no difference in burden between male *Rag2*$^{-/-}$ and *Rag2*$^{+/-}$ controls (**Figure 4A**). However, female *Rag2*$^{-/-}$ mice had significantly increased MRSA burden compared to *Rag2*$^{+/-}$ littermates (**Figure 4A**). These data suggest that a B or T cell-mediated response is required for the sex bias we observe in colonization resistance.

B cell associated *Ighv* genes were differentially regulated between males and females. However, *Ighm*$^{-/-}$ (μMT) mice, which lack mature B lymphocytes, were similar to *Ighm*$^{+/-}$ controls and retained sex differences; males displayed prolonged colonization and females were resistant (**Figure 4B**). In contrast, antibody-mediated depletion of CD4+ T cells (**Figure 4—figure supplement 1D**) substantially increased MRSA GI colonization of female mice, similar to *Rag2*$^{-/-}$ mice (**Figure 4C**). Therefore, CD4+ T cells are required for MRSA colonization resistance associated with female sex.

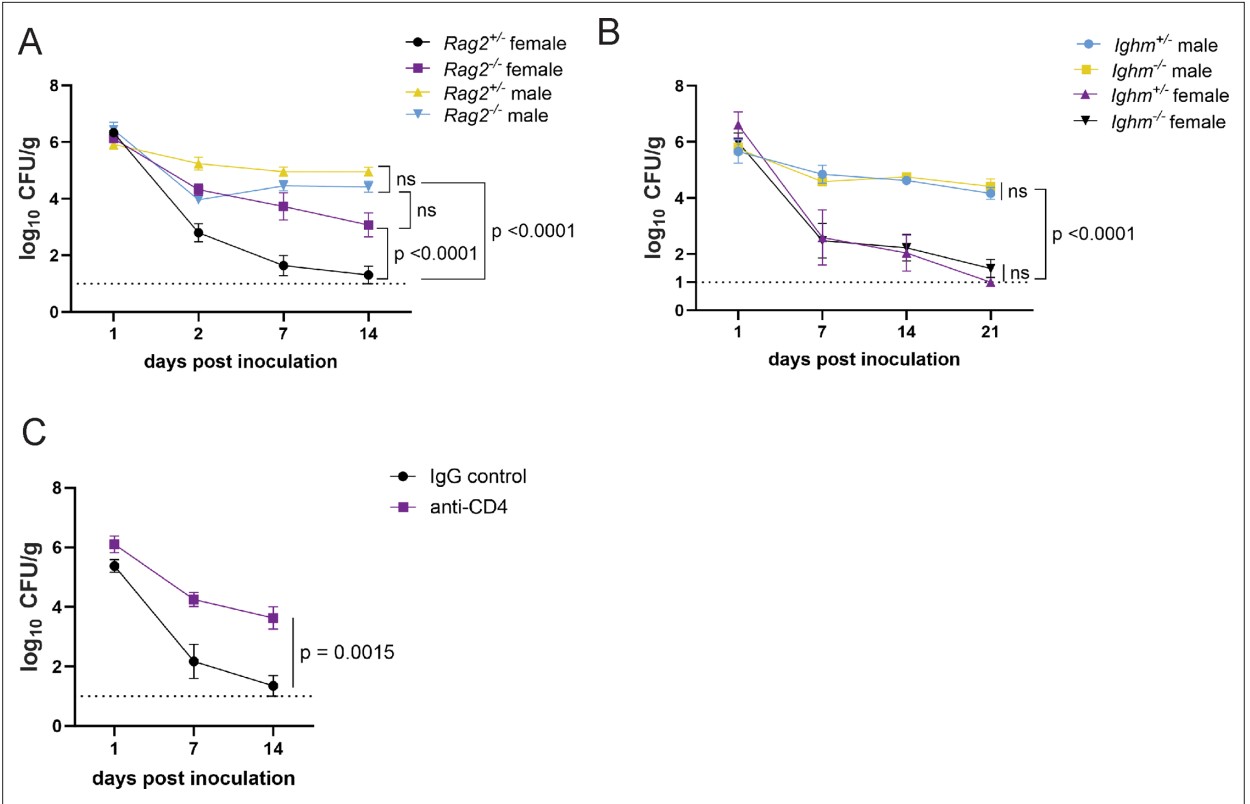

**Figure 4.** CD4+ T cells mediate colonization resistance in female mice. (**A**) Methicillin-resistant *S. aureus* (MRSA) colony forming units (CFU) in stool following oral inoculation of *Rag2⁻/⁻* and *Rag2⁺/⁻* mice bred at NYU. Male *Rag2⁻/⁻* n=12, male *Rag2⁺/⁻* n=12, female *Rag2⁻/⁻* n=12, female *Rag2⁺/⁻* n=11. (**B**) MRSA CFU in stool following oral inoculation of *Ighm⁻/⁻* and *Ighm⁺/⁻* mice bred at NYU. Male *Ighm⁻/⁻* n=6, male *Ighm⁺/⁻* n=6, female *Ighm⁻/⁻* n=8, female *Ighm⁺/⁻* n=5. (**C**) MRSA CFU in stool following oral inoculation of female NYU B6 mice injected intraperitoneally (IP) with 250 µg of anti-CD4 depleting antibody or anti-IgG control. Anti-IgG n=8, anti-CD4 n=10. Data points represent mean ± SEM from at least two independent experiments. Statistical analysis: area under the curve analyzed by a one-way ANOVA with Sidak's multiple comparison test for (**A**) and (**B**) and a two-tailed t-test for (**C**). ns: not significant.

The online version of this article includes the following figure supplement(s) for figure 4:

**Figure supplement 1.** NLRP3 inflammasome activation is not required for methicillin-resistant *S. aureus* (MRSA) colonization resistance in female NYU mice.

## MRSA colonization resistance in female mice is dependent on Th17 cells and neutrophils

In addition to T cells, our transcriptome analysis showed upregulation of genes associated with neutrophil function (*Figure 3B*). In mouse models of nasal colonization, clearance is mediated by neutrophil influx downstream of the type 17 response, which includes interleukin-17A (IL-17A) producing lymphoid cells such as Th17 cells (*Archer et al., 2013*; *Archer et al., 2016*). As an extracellular Gram-positive bacterium at the mucosal surface in the gut, MRSA may be subject to regulation by preexisting Th17 cells (*Nagashima et al., 2023*; *Bacher et al., 2019*; *Cassotta et al., 2021*). Although the total numbers of CD4+ T cells were similar across conditions, we observed an increase in IL-17A+ CD4+ T cells in the small intestine and colon of female mice 2 dpi with MRSA that was not observed in males (*Figure 5A and B*, *Figure 5—figure supplement 1A and B*, and *Figure 5—figure supplement 2A and B*). There were no differences in γδ T cells and innate lymphoid cells or the proportion of these lymphoid subsets that were IL-17A+ (*Figure 5C and D* and *Figure 5—figure supplement 2C and D*). To determine the importance of the type 3 immune response that encompasses Th17 cells, we inoculated *RAR-related orphan receptor gamma* (*Rorc*) deficient mice that lack the transcriptional regulator required for the differentiation of these cell types. Although there was no difference in MRSA burden between male *Rorc ⁺/⁻* and *Rorc⁻/⁻* mice that remained colonized, female *Rorc⁻/⁻* mice had higher MRSA burden compared to *Rorc ⁺/⁻* controls (*Figure 5E*). Responses to IL-17 itself was

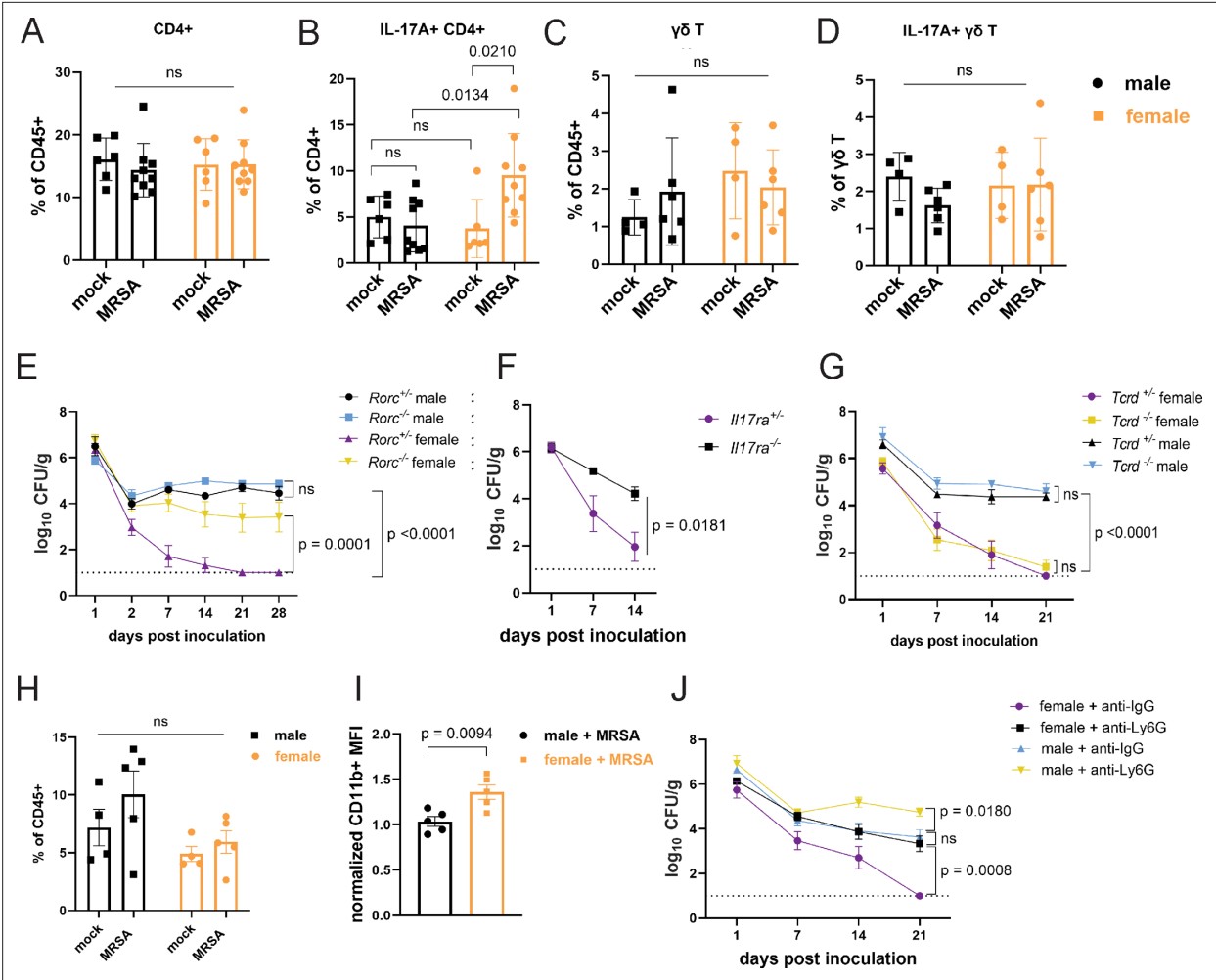

**Figure 5.** Female mice have increased IL-17A+ CD4+ T cells and require neutrophils for methicillin-resistant *S. aureus* (MRSA) clearance. (**A**) Flow cytometry of cecal-colonic lamina propria CD4+ T cells as a percentage of CD45+ cells in male and female NYU mice treated with phosphate-buffered saline (PBS) or MRSA 2 days post inoculation (dpi). (**B**) Flow cytometry of cecal-colonic lamina propria IL17A+ CD4+ T cells as a percentage of total CD4+ T cells in male and female NYU mice treated with PBS or MRSA 2 dpi. (**C**) Flow cytometry of cecal-colonic lamina propria γδ T cells as a percentage of CD45+ cells in male and female NYU mice treated with PBS or MRSA 2 dpi. (**D**) Flow cytometry of cecal-colonic lamina propria IL17A+ γδ T cells as a percentage of total CD4+ T cells in male and female NYU mice treated with PBS or MRSA 2 dpi. (**E**) MRSA colony forming units (CFU) in stool following oral inoculation of *Rorc*$^{-/-}$ and *Rorc*$^{+/-}$ mice bred at NYU. Male *Rorc*$^{+/-}$ n = 5, male *Rorc*$^{-/-}$ n=7, female *Rorc*$^{+/-}$ n = 9, female *Rorc*$^{-/-}$ n=9. (**F**) MRSA CFU in stool following oral inoculation of female *Il17ra*$^{+/-}$ and *Il17ra*$^{-/-}$ mice bred at NYU. *Il17ra*$^{+/-}$ n = 6, *Il17ra*$^{-/-}$ n=6. (**G**) MRSA CFU in stool following oral inoculation of *Tcrd*$^{+/-}$ and *Tcrd*$^{-/-}$ mice bred at NYU. Male *Tcrd*$^{+/-}$ n = 6, male *Tcrd*$^{-/-}$ n=6, female *Tcrd*$^{+/-}$ n=8, and female *Tcrd*$^{-/-}$ n=12. (**H**) Flow cytometry of cecal-colonic lamina propria Ly6G+CD11b+ neutrophils as a percentage of CD45+ cells in male and female NYU mice treated with PBS or MRSA 2 dpi. (**I**) Quantification of mean fluorescence intensity (MFI) of surface CD11b on neutrophils by flow cytometry normalized to mock-treated controls. (**J**) MRSA CFU in stool following oral inoculation of NYU B6 mice treated with anti-Ly6G neutrophil depleting antibody or anti-IgG control. Male anti-IgG n=6, male anti-Ly6G n=8, female anti-IgG n = 12, and female anti-CD4+ n = 15. Data points represent mean ± SEM from at least two independent experiments. Statistical analysis: two-way ANOVA+Sidak's multiple comparisons test for (**A**–**D**) and (**H**), area under the curve followed by a one-way ANOVA+Sidak's multiple comparisons test for (**E**), (**G**), (**J**), or a two-tailed t-test for (**F**) and a two-tailed t-test for (**I**). ns: not significant.

The online version of this article includes the following figure supplement(s) for figure 5:

**Figure supplement 1.** Flow cytometry gating scheme for lymphoid subsets in the lamina propria.

**Figure supplement 2.** Immune cell populations detected by flow cytometry.

required, because female mice deficient in the IL-17 receptor (*IL-17ra*) had increased MRSA burden compared to heterozygous littermates (*Figure 5F*). There was no difference in MRSA burden between *Tcrd*$^{-/-}$ and *Tcrd*$^{+/-}$ male or female mice (*Figure 5G*), confirming that γδ T cells were not required for colonization resistance. Taken together, these data implicate Th17 cells and IL-17 in colonization resistance against MRSA observed in female mice.

Th17 cells and IL-17 can recruit and promote antimicrobial functions of neutrophils (*McGeachy et al., 2019*). Neutrophils from female mice and humans display signs of increased maturity and activation (*Blazkova et al., 2017*; *Er-Lukowiak et al., 2023*; *Gupta et al., 2020*; *Lu et al., 2021*). Although we did not observe significant differences in the number of neutrophils between conditions (*Figure 5H*), neutrophils from female mice 2 dpi with MRSA had increased levels of CD11b, a sign of activation (*Parkos et al., 1994*), compared to males (*Figure 5I*). Antibody-mediated depletion of neutrophils (*Figure 5—figure supplement 2E*) led to increased MRSA burden compared with isotype control-treated mice (*Figure 5J*). Although colonization was increased in males, neutrophil depletion in female mice led to an earlier and more pronounced increase in MRSA colonization that was prolonged. These results are consistent with the RNA-seq analysis suggesting that a neutrophil response occurs in both sexes. It is likely that neutrophils control bacterial burden to some degree but fail to promote clearance in males. In contrast, neutrophils are required for clearance of MRSA in the gut of female mice.

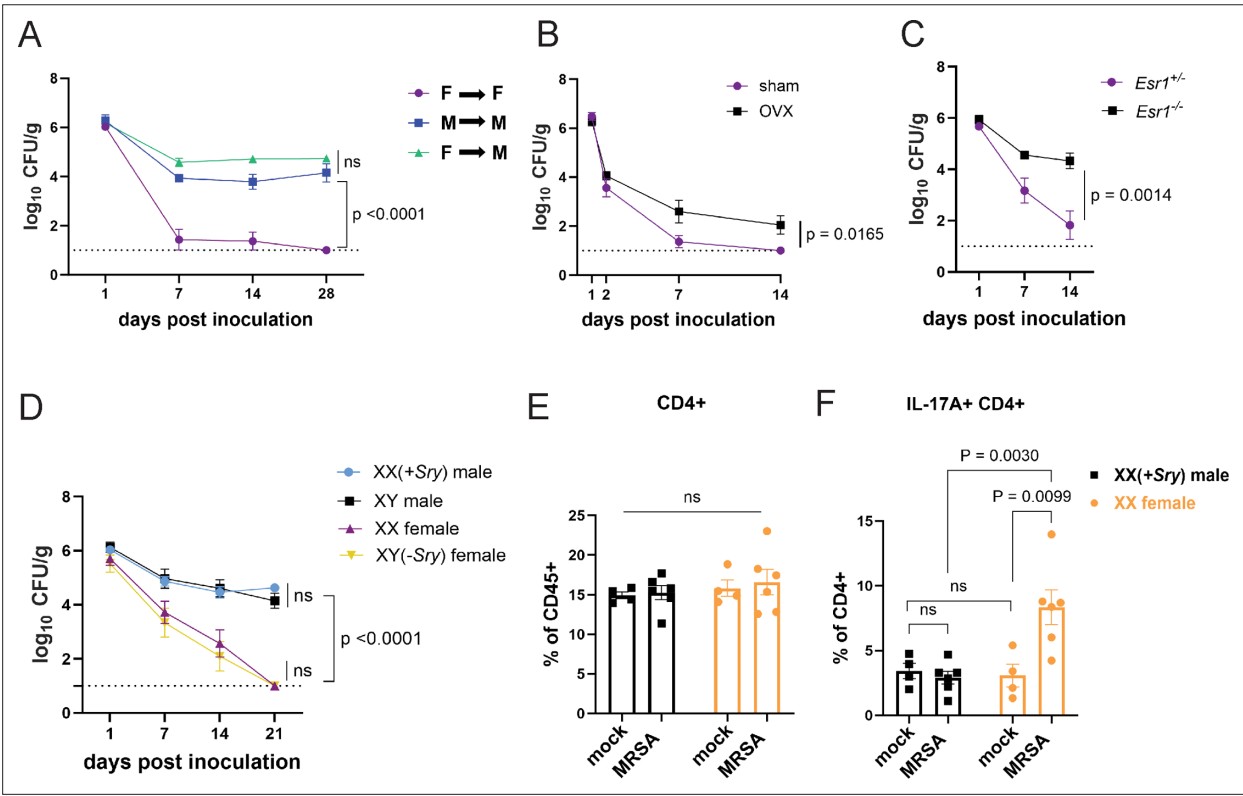

**Figure 6.** Sex hormones, not sex chromosomes, mediate methicillin-resistant *S. aureus* (MRSA) colonization resistance in female mice. (**A**) MRSA colony forming units (CFU) in stool following oral inoculation of male or female mice that were irradiated and reconstituted with bone marrow (BM) from donor male or female mice. Female BM into female recipients (F→F) n=8, male BM into male recipients (M→M) n=7, female BM into male recipients (F→M) n=8. (**B**) MRSA CFU in stool following oral inoculation of ovariectomized female mice or sham operated littermate controls. Ovariectomized (OVX) n=10, sham n=10. (**C**) MRSA CFU in stool following oral inoculation of *Esr1*[+/-] and *Esr1*[-/-] female mice bred at NYU. *Esr1*[+/-] n = 6, *Esr1*[-/-] n=6. (**D**) MRSA CFU in stool following oral inoculation of four core genotype mice. XX n=5, XY(-Sry) n=5. XY n=5, XX(+Sry) n=4. (**E**) CD4+ T cells from cecal-colonic lamina propria of XX females and XX(+Sry) males 2 days post inoculation (dpi) inoculation with MRSA or mock control. (**F**) Percentage of IL-17A+CD4+ T cells in cecal-colonic lamina propria of XX females and XX(+Sry) males 2 dpi inoculation with MRSA or mock control. Data points represent mean ± SEM from at least two independent experiments. Statistical analysis: area under the curve followed by a one-way ANOVA with Sidak's multiple comparisons for (**A**, **C**) and two-tailed t-test for (**B**, **F**) and two-way ANOVA+Sidak's multiple comparisons test for (**D–E**). ns: not significant.

The online version of this article includes the following figure supplement(s) for figure 6:

**Figure supplement 1.** Sex hormone receptor expression in cecal-colonic lamina propria immune and epithelial cells.

# Sex hormones, not sex chromosomes, mediate MRSA colonization resistance in female mice

Sex bias in immunity could be mediated by genes that are encoded on sex chromosomes. Many X linked genes are involved in immunity, such as pattern recognition receptors, and incomplete X inactivation can lead to higher expression of such genes (*Schurz et al., 2019*). To test whether colonization resistance was due to sex chromosomes within a hematopoietic cell type, we generated chimeras in which wild-type male mice were reconstituted with T-lymphocyte depleted bone marrow (BM) from female donors and compared with male and female mice that received BM from same sex donors. The reciprocal chimera in which female mice are reconstituted with male BM was not feasible due to transplant rejection. Although female mice that received female donor BM were resistant to MRSA, male mice that received male or female donor BM remained colonized (*Figure 6A*). Thus, it is unlikely that a gene expressed on a sex chromosome in immune cells such as T cells mediates colonization resistance in females, raising the possibility of a role for sex hormones in our model. The hormones estrogen, progesterone, and testosterone are known to modulate trafficking and function of immune cells. For example, estrogen can enhance responses to extracellular pathogens (*Fuseini et al., 2019*; *Yasuda et al., 2019*; *Békési et al., 2001*). To test whether sex hormones are involved in GI colonization resistance to MRSA, female mice at 6 weeks of age were ovariectomized (OVX) or given a sham operation (SO) and allowed to recover for 2 weeks prior to oral inoculation with MRSA. OVX mice had increased MRSA burden and duration of carriage compared to SO controls (*Figure 6B*). Estrogen receptor-α (ERα, encoded by *Esr1*) signaling increases Th17 cell proliferation and production of IL-17 (*Fuseini et al., 2019*; *Newcomb et al., 2015*). We confirmed *Esr1* was expressed in cecal-colonic immune cells, and that it displayed a nonsignificant trend suggesting an increase upon MRSA inoculation (*Figure 6—figure supplement 1A*). Female *Esr1*[-/-] mice had increased MRSA burden compared to heterozygous littermates (*Figure 6C*), indicating that estrogen mediates the enhanced colonization resistance in female mice.

To formally decouple the role of sex hormones and sex chromosome encoded gene products, we used the mouse four core genotype (FCG) model in which the gene sex determining region Y (*Sry*) that defines male sex has been moved from the Y chromosome to an autosome (*Arnold and Chen, 2009*). Thus, the sex chromosome complement (XX or XY) does not relate to gonadal sex in the FCG model. We found that XY(-*Sry*) gonadal female mice lose MRSA carriage analogous to XX females, consistent with prior studies showing that they display similar estradiol levels (*Palaszynski et al., 2005*; *Sasidhar et al., 2012*), and both XY and XX(+*Sry*) gonadal male mice remain persistently colonized at similar levels (*Figure 6D*). Like their wild-type counterparts, we do not observe a change in the proportions of CD4+ T cells in the lamina propria between XX males and XX females with and without MRSA (*Figure 6E*). Instead, XX(-*Sry*) females displayed an increase in IL17A+CD4+ T cells in the lamina propria compared with XX(+*Sry*) males 2 dpi MRSA inoculation (*Figure 6F*). These findings further support a hormone-mediated effect on the Th17 cell response in female mice rather than a chromosomal one.

## Discussion

*S. aureus* intestinal carriage is common and associated with infection, but how variables such as sex contribute to colonization susceptibility have been obscure. We established a mouse model to investigate MRSA intestinal colonization, which revealed a sex-specific effect in which female mice rapidly cleared MRSA, while their male counterparts remained colonized. This protection was microbiota dependent because mice lacking a microbiota or those with a less diverse microbiota were susceptible to persistent colonization. Microbiota composition alone, however, was insufficient to explain the sex-dependent colonization resistance observed. Females displayed an enhanced immune response to MRSA colonization characterized by increases in Th17 cells and neutrophil activation. Increase in MRSA burden in OVX or ERα-deficient females and XX (+*Sry*) gonadal male mice indicate that this effect of the female sex is hormonally mediated rather than dependent on genes present on sex chromosomes. Collectively, our results support a model in which GI colonization resistance against MRSA in female mice is dependent on the microbiota and an enhanced Th17 response downstream of sex hormones.

Sex steroid hormones have well-documented effects on the immune response, including CD4+ T cells (*Fuseini et al., 2019*; *Chi et al., 2024*; *Li et al., 2024*; *Taneja, 2018*; *Karpuzoglu-Sahin et al., 2001a*; *Karpuzoglu-Sahin et al., 2001b*). Our results are consistent with studies showing that ERα signaling increases differentiation and cytokine production of Th1 and Th17 cells (*Fuseini et al., 2019*; *Mohammad et al., 2018*; *Maret et al., 2003*), although higher levels of estrogen characteristic of pregnancy can stimulate immunosuppressive regulatory CD4+ T cell conversion (*Tai et al., 2008*). Recent work identified a parallel role for the male sex hormone androgen in suppressing lymphoid and neutrophil responses during intradermal infection of mice with *S. aureus* (*Chi et al., 2024*; *Li et al., 2024*). In this context, introduction of a complex microbiota into GF mice amplified the skin type 17 sex bias in females (*Chi et al., 2024*). Sex biases in neutrophil function, including increased phagocytosis and extracellular trap formation in female mice, have been observed (*Castleman et al., 2018*; *Yasuda et al., 2019*; *Spitzer and Zhang, 1996*; *Pokhrel et al., 2020*). In line with our findings of increased CD11b on intestinal neutrophils isolated from female mice exposed to MRSA, studies examining MRSA skin and soft tissue infections found that BM neutrophils from female mice have an enhanced ability to kill MRSA ex vivo compared to those from male mice, which was linked to increases in surface CD11b and antimicrobial production (*Castleman et al., 2018*; *Pokhrel et al., 2020*).

Although sex difference was not reported, colonization resistance against *S. aureus* in the nares is also associated with Th17 cells and neutrophils (*Archer et al., 2013*; *Archer et al., 2016*), supporting the general importance of the CD4+ T cell response over antibodies and B cells in determining susceptibility to colonization. The absence of a sterilizing B cell response may reflect immune evasion strategies such as the bacterially produced Protein A that binds to the Fcγ portion of immunoglobulins, protecting *S. aureus* from opsonophagocytic killing (*Falugi et al., 2013*; *Pauli et al., 2014*). Our finding that female mice lacking mature B cells were still able to resist MRSA GI colonization is mirrored by a human volunteer study demonstrating that antibodies prior to intranasal inoculation did not block persistent colonization by their cognate *S. aureus* strains and that antibody levels did not distinguish intermittent and noncarriers (*van Belkum et al., 2009*). In contrast, low CD4+ T cell counts in HIV+ individuals are a risk factor for *S. aureus* nasal colonization (*de Ferreira et al., 2014*), and rates of colonization were higher in HIV+ males compared to HIV+ females (*Neupane et al., 2018*). Understanding the cellular response to MRSA colonization and the impact of host sex may inform vaccination strategies (*Brown et al., 2014*).

We observed an increase in IL-17A+ CD4+ T cells in colonization resistant females at 2 dpi with MRSA, a duration that is typically insufficient for a de novo antigen-specific adaptive immune response. It is likely that the gut harbors pre-existing T cells that can rapidly respond. The microbiota dependence of colonization resistance provides a potential explanation for this observation. Superantigens from *S. aureus* bind directly to Vβ regions of T cell receptors (TCRs) and MHC class II on antigen-presenting cells, resulting in hyperactivation of T lymphocytes and monocytes/macrophages. *S. aureus* superantigens induce a robust IL-17 response from memory Th17 cells from adult humans but not from naive T cells (*Islander et al., 2010*), raising the possibility that such a mechanism can act on Th17 cells selected by the microbiota. Also, CD4+ T cells in the gut, including Th17 cells, can recognize antigens that are shared by taxonomically diverse bacteria in the gut (*Nagashima et al., 2023*). In the colon of healthy humans and mice, MHC-II restricted CD4+ T cells with Th17 functionality have been identified that are responsive to commensal microbial antigens in an innate-like manner (*Hackstein et al., 2022*). Additionally, intestinal colonization by the fungus *Candida albicans* favors Th17 polarization associated with increased IL-17 and neutrophils that protect mice from intravenous infection with MRSA (*Shao et al., 2019*; *Chen et al., 2023*). Thus, it is possible that we are observing either a cross reactive Th17 response or a bystander effect initiated by a commensal.

Investigating the antigen-specific response to MRSA is challenging due to bacterially encoded superantigens and other secreted virulence factors, such as α-hemolysin and LukED, that induce cell death of lymphocytes (*Tam and Torres, 2019*). Therefore, the relationship between the microbiota and sex-dependent colonization resistance remains unanswered in our model. Other possibilities include microbiota-mediated regulation of estrogen and its receptor or signaling. However, given the strong connection between microbiota and Th17 (*Ivanov et al., 2009*; *Atarashi et al., 2015*), we propose a speculative model in which initial microbiota differences prime the gut for a Th17 and neutrophil response to MRSA that is enhanced by estrogen. Identifying the microbe(s) that are necessary for

colonization resistance in females and careful examination of the dynamic regulation of sex hormones in this model will be insightful.

Animal models that incorporate sex as a variable are critical to conduct rigorous, translational science and build toward personalized medicine. Our study reveals an interplay between host microbiota, immune response, and sex steroid hormones in response to intestinal exposure to MRSA, a common commensal and major source of life-threatening invasive infections. Given the importance of the gut as a site of colonization and transmission for many medically important infectious agents, we suggest careful documentation of sex-specific effects in future studies examining this host-microbe interface.

# Materials and methods

## Mice

Mice designated as JAX refer to male and female 6- to 8-week-old C57BL/6J mice purchased from Jackson Laboratory and used directly for experiments. NYU C57BL/6J breeders were originally purchased from Jackson Laboratory and bred onsite at New York University Grossman School of Medicine to generate littermate male and female mice for comparison. Every 6 months breeders were replaced by using a new male from Jackson Laboratory to pair with a NYU female to reduce genetic drift. $Rag2^{-/-}$, $Tcrd^{-/-}$, $Ighm^{-/-}$, $Ptprc^a$ (B6 CD45.1), and $Esr1^{tm1Ksk+/-}$ mice were purchased from Jackson Laboratory. $Rorc(\gamma t)$-enhanced GFP ($Rorc^{-/-}$) mice were previously described (**Chen et al., 2023**). $Il17ra^{-/-}$ mice were generated by Amgen Inc (Seattle, WA, USA) and provided by Dr. Jeffrey Weiser. Littermate heterozygous controls for $Rag2^{-/-}$, $Ighm^{-/-}$, and $Rorc^{-/-}$ mice were generated by initially crossing homozygous knockout mice with wild-type C57BL/6J mice to establish heterozygotes that were then used to generate homozygous and heterozygous breeder pairs.

GF C57BL/6J were bred and maintained in flexible-film isolators at the New York University Grossman School of Medicine Gnotobiotics Animal Facility (**Sargsian et al., 2022**; **Dallari et al., 2021**; **Kernbauer et al., 2014**). Absence of fecal bacteria was confirmed monthly by evaluating the presence of 16S rDNA in stool samples by qPCR. Minimal flora mice harboring the consortium of 15 bacteria (Oligo-MM$_{12}$+FA3) (**Brugiroux et al., 2016**) were maintained in a separate isolator as previously described (**Sargsian et al., 2022**; **Dallari et al., 2021**). For inoculation with bacteria, GF mice and minimal flora mice were housed in Bioexclusion cages (Tecniplast) with access to sterile food and water.

The sample size for animal experiments was chosen based on previous data generated in the laboratory. All animal studies were performed according to protocols approved by the NYU Grossman School of Medicine Institutional Animal Care and Use Committee.

## MRSA intestinal colonization in mice

Prior to inoculation, stool from C57BL/6J mice was homogenized and plated on CHROMagar *Staph aureus* (CHROMagar, Paris, France) selective plates to ensure mice were free of *S. aureus* carriage. C57BL/6J mice were orally gavaged with ~1 × 10$^8$ CFU of CA-MRSA strain USA300 (LAC). Stool samples were collected from each mouse on indicated days post inoculation. Screw-cap tubes (2 mL) filled with 1.0 mm beads were weighed before and after the addition of stool to determine weight. Sterile phosphate-buffered saline (PBS) (1 mL) was added to each tube, which were vigorously shaken in a bead-beater (MP Biomedicals, Santa Ana, CA, USA) for 60 s. Stool aliquots were diluted and plated to enumerate viable bacteria. Samples were plated on CHROMagar MRSA II (BBL)/Mannitol salt agar plates, incubated at 37°C for 24 hr, and colonies were counted to determine MRSA burden per gram of stool.

## Intestinal lamina propria and epithelial cell isolation

Colonic and cecal tissues were flushed with HBSS (Gibco), fat and Peyer's patches were removed, and the tissue was cut into 6–8 pieces. Tissue bits were incubated first with 20 mL of HBSS with 2% HEPES (Corning), 1% sodium pyruvate (Corning), 5 mM EDTA, and 1 mM dithiothreitol (Sigma-Aldrich) for 15 min at 37°C with shaking, and then with new 10 mL of HBSS with 2% HEPES, 1% sodium pyruvate, 5 mM EDTA for 10 min at 37°C with shaking. The samples were filtered by 40 μm cell strainer (BD) and the supernatant containing intestinal epithelial cells was collected and subjected to gradient

centrifugation using 40% and 80% Percoll (Sigma-Aldrich). The upper layer containing epithelial cells between the 40% and 80% gradients was collected. Tissue bits were washed in HBSS+5% FCS, minced, and then enzymatically digested with collagenase D (0.5 mg/mL, Roche) and DNase I (0.01 mg/mL, Sigma-Aldrich) for 30 min at 37°C with shaking. Digested solutions were passed through a 40 mm cell strainer and cells were subjected to gradient centrifugation using 40% and 80% Percoll (Sigma-Aldrich). The lower layer containing immune cells between the 40% and 80% gradients was collected.

## Flow cytometry

Lamina propria cells from either cecal and colonic or small intestinal tissue were harvested as described. For intracellular cytokine staining, cells were stimulated using the eBioscience cell stimulation cocktail for 4 hr at 37°C. The cells were fixed and permeabilized using the BioLegend fixation and permeabilization buffer. The following antibodies (clones) were used for staining: CD45 (30-F11), Ly6G (1A8), TCR-$\beta$ (H57-597), CD4 (GK1.5), CD8 (53–6.7), IL-17A (TC11-18H10.1), TCR $\gamma/\delta$ (GL3), CD127(SB/199), and CD11b (M1/70). All samples were blocked with Fc Block (TruStain FcX). Zombie UV Fixable Viability Kit (BioLegend) was used to exclude dead cells prior to gating for other markers. Samples were run on BD FACS Symphony A5 and FACsFlowJo v.10 was used to analyze the flow cytometry data.

## Antibody-mediated depletion experiments

C57BL/6J mice bred at NYU were injected intraperitoneally with either 200 µg rat anti-mouse Ly6G or rat IgG2a isotype control antibody to deplete neutrophils and 250 µg rat anti-mouse CD4 or rat IgG2a isotype control antibody to deplete CD4+ T cells (Bio-X-Cell, West Lebanon, NH, USA). Anti-Ly6G injections occurred 1 day prior to MRSA inoculation and then every 3 days until mice were sacrificed 14 dpi. Anti-CD4 injections occurred 3 days prior to inoculation and every 7 days after initial injection until mice were sacrificed 14 dpi.

## BM chimera experiments

Eight-week-old recipient CD45.1 congenic C57BL/6J mice bred at NYU received 550 rads in 2 doses over 2 sequential days and were injected retro-orbitally with 2×10$^6$ T lymphocyte-depleted BM cells from either female or male CD45.2 congenic C57BL/6J donors bred at NYU. Mature T lymphocytes were depleted from BM cell suspension using the CD3$\epsilon$ MicroBead Kit (Miltenyi Biotech). Mice were allowed 8 weeks for reconstitution before oral inoculation with 1×10$^8$ CFU MRSA. BM reconstitution of the CD45+ compartment in CD45.1 mice was confirmed by flow cytometry analysis of CD45.2+ cells in the BM at the time of sacrifice.

## Ovariectomy surgery

Six-week-old female mice bred at NYU were anesthetized with isoflurane and placed in ventral recumbency with tail toward surgeon. Ophthalmic ointment was applied bilaterally and heat support was provided throughout the procedure. The dorsal mid-lumbar area was shaved in an area 150% greater than anticipated incision length and swabbed three times with alternating scrubs of betadine and alcohol. A 1–1.5 cm dorsal midline skin incision was made halfway between the caudal edge of the ribcage and the base of the tail. A single incision of 5–7 mm long was made into the muscle wall on both the right and left sides approximately one-third of the distance between the spinal cord and the ventral midline. The ovary and the oviduct were exteriorized through the muscle wall. A hemostat was clamped around the uterine vasculature between the oviduct and uterus. Each ovary and part of the oviduct was removed with single cuts through the oviducts near the ovary. The hemostat was removed and the remaining oviduct was assessed for hemorrhage. Hemostasis was confirmed prior to placing the remaining tissue into the peritoneal cavity. The ovary on the other side was removed in a similar manner. The muscle incision was closed with monofilament absorbable suture in a cruciate pattern. Bupivacaine was applied on the closed muscle layer. The skin incision was closed in a cruciate pattern with sterile skin sutures (monofilament nonabsorbable) and a small amount of tissue glue. The skin sutures were removed 10 days after surgery. Sham surgery mice underwent the same surgery but did not have their ovaries removed. Mice were administered Carprofen SQ every 24 hr for 3 days following the procedure. Mice were allowed 2 weeks to fully recover before oral inoculation with 1×10$^8$ CFU of MRSA.

## RNA sequencing

RNA was extracted from lamina propria cells and epithelial fraction isolated from the cecum-colon using RNeasy Plus Mini Kit (QIAGEN). Four male and four female B6 mice bred at NYU were used for each experimental condition. An RNA library was prepared using the Illumina TruSeq RNA sample preparation kit and sequenced with the Illumina HiSeq 2000 using the TruSeq RNA v.2 protocol. Illumina CASAVA v.1.8.2 was used to generate FASTQ files containing 29.5–53.5 million qualified reads per sample. Alignment and gene expression count were computed using default settings, which aligns reads to the union of all RefSeq-annotated exons for each gene.

RNA-seq results were processed using the v.4 R package DESeq2 v.3 to obtain variance stabilized count reads, fold changes relative to specific condition, and statistical p-value. Analysis of the transcriptome focused on differentially expressed genes (DEGs), defined as the genes with an absolute log2 fold change relative to specific condition >1.2 and an adjusted p-value<0.05. Enriched pathways for the DEGs were analyzed by Ingenuity Pathway Analysis (QIAGEN). The analyses were visualized using R package ggplot2 (*Wickham, 2016*).

## DNA extraction from stool

DNA were extracted from stool using the MagMAX Microbiome Ultra Nucleic Acid Isolation Kit (Thermo Fisher) following the manufacturer's instructions with modifications. An initial bead-beating step using differentially sized beads (glass beads, 0.5–0.75 mm; zirconia beads, <100 μm) was included and lysozyme (20 mg/mL) was added to the lysis buffer.

## 16S library preparation and sequencing analysis

Bacterial 16S rRNA genes were amplified at the V4 region using 16S universal primer pairs and amplicon sequencing was performed on the Illumina MiSeq system, yielding 150 bp paired reads.

16S Amplicon PCR Forward Primer = 5' TCGTCGGCAGCGTCAGATGTGTATAAGAGACAGCCT ACGGGNGGCWGCAG, 16S Amplicon PCR Reverse Primer = 5' GTCTCGTGGGCTCGGAGATGTGTA TAAGAGACAGGACTACHVGGGTATCTAATCC.

The sequencing reads were processed using the DADA2 v.1 pipeline in the QIIME2 v.2022.2 software package. The cloud based platform Nephele v.2 was used to run QIIME2 analysis (*Weber et al., 2018*). The minimum Phred quality score of 20 was applied to ensure high-quality sequence data. An open-reference clustering algorithm was used to identify OTUs based on a 97% similarity threshold to a reference database. Chimera removal was conducted to eliminate artifacts in the data. Taxonomic classification was performed using a search-based method to assign taxonomy to the OTUs. Barplots were generated with a minimum frequency filter of 1000 to visualize the abundance of microbial taxa. A percentage identity threshold of 97% was used to assign taxonomy at the genus level using the SILVA rRNA database. PCoA and calculation of Shannon index were performed using the phyloseq R package (*McMurdie and Holmes, 2013*) version 1.46 after rarefaction to a depth of 1000.

## Statistical analysis

The number of animals per group is annotated in corresponding figure legends. GraphPad Prism v.10 was used to generate graphs and assess significance for bacterial burden and weight loss, and flow cytometry data were analyzed using FlowJo v.10. The MRSA burden curves were analyzed using area under the curve followed by a one-way ANOVA with Sidak's multiple comparisons test when comparing more than two experimental conditions and a two-tailed t-test when comparing two experimental conditions. An unpaired two-tailed Student's t-test was used to evaluate the differences between two groups. Welch's correction was used when variances were significantly different between groups. A two-way ANOVA with Sidak's multiple comparisons test was used to evaluate experiments involving multiple groups. All p-values are shown in the figures.

## Acknowledgements

We would like to thank Margie Alva, Juan Carrasquillo, and David Basnight for their help in the NYU Gnotobiotic Facility, the NYU Flow Cytometry Core for training and access to equipment, the NYU Genome Technology Center for processing and sequencing of 16S and RNA samples, the NYU Experimental Pathology Research Laboratory for processing of H&E tissue samples, and the NYU Reagent

Preparation service for providing bacterial media and plates. Core facilities were supported by NIH grant P31CA016087. We would like to thank Dr. Mariya London for her help with flow cytometry technique and analysis. We would also like to thank members of the Cadwell, Shopsin, and Torres Labs for their constructive comments. This work was supported in part by NIH grants DK093668 (KC), AI121244 (KC, VJT), HL123340 (KC), AI130945 (KC), AI140754 (BS, VJT, KC), AI179896 (KC), DK 050306 (KC), and DK124336 (KC) and NIH grant 2T32AI100853-11 (AL). We would like to acknowledge the Vilcek Institute of Graduate Biomedical Sciences for their support.

## Additional information

### Competing interests

Victor J Torres: has received honoraria from Pfizer and MedImmune and is an inventor on patents and patent applications (US8431,687B2; US2019135900-A1; EP4313303A1) filed by New York University, which are currently under commercial license to Janssen Biotech Inc Janssen Biotech Inc provides research funding and other payments associated with a licensing agreement. Ken Cadwell: has received research support from Pfizer, Takeda, Pacific Biosciences, Genentech, and Abbvie. Has consulted for or received an honoraria from Puretech Health, Genentech, and Abbvie. Is an inventor on U.S. patent 10,722,600 and provisional patent 62/935,035 and 63/157,225. The other authors declare that no competing interests exist.

### Funding

| Funder | Grant reference number | Author |
| --- | --- | --- |
| National Institutes of Health | AI140754 | Victor J Torres<br>Bo Shopsin<br>Ken Cadwell |
| National Institutes of Health | DK093668 | Ken Cadwell |
| National Institutes of Health | AI121244 | Ken Cadwell<br>Victor J Torres |
| National Institutes of Health | HL123340 | Ken Cadwell |
| National Institutes of Health | AI130945 | Ken Cadwell |
| National Institutes of Health | AI179896 | Ken Cadwell |
| National Institutes of Health | DK050306 | Ken Cadwell |
| National Institutes of Health | DK124336 | Ken Cadwell |
| National Institutes of Health | 2T32AI100853-11 | Alannah Lejeune |

The funders had no role in study design, data collection and interpretation, or the decision to submit the work for publication.

### Author contributions

Alannah Lejeune, Data curation, Formal analysis, Visualization, Writing – original draft, Writing – review and editing; Chunyi Zhou, Data curation, Supervision, Writing – review and editing; Defne Ercelen, Formal analysis, Visualization, Writing – review and editing; Gregory Putzel, Software, Formal analysis, Visualization, Writing – review and editing; Xiaomin Yao, Miranda Pawline, Data curation; Alyson R Guy, Methodology, Writing – review and editing; Magdalena Podkowik, Data curation, Methodology, Writing – review and editing; Alejandro Pironti, Visualization, Writing – review and editing; Victor J Torres, Conceptualization, Funding acquisition; Bo Shopsin, Conceptualization, Resources, Supervision, Funding acquisition, Investigation, Writing – review and editing; Ken Cadwell, Conceptualization,

Resources, Software, Supervision, Funding acquisition, Investigation, Writing – original draft, Project administration, Writing – review and editing

### Author ORCIDs
Victor J Torres ⓘD https://orcid.org/0000-0002-7126-0489
Bo Shopsin ⓘD https://orcid.org/0009-0001-7729-8584
Ken Cadwell ⓘD https://orcid.org/0000-0002-5860-0661

### Ethics
All animal studies were performed according to protocols approved by the NYU Grossman School of Medicine Institutional Animal Care and Use Committee,IA16-01941.

Reviewer #1 (Public review): https://doi.org/10.7554/eLife.101606.3.sa1
Reviewer #3 (Public review): https://doi.org/10.7554/eLife.101606.3.sa2
Author response https://doi.org/10.7554/eLife.101606.3.sa3

---

## Additional files

### Supplementary files
MDAR checklist

### Data availability
Sequencing data have been deposited in NCBI Sequence Read Database (SRA). The sequencing accession number for the bulk RNA sequencing is PRJNA1134782 and the accession number for the 16S rRNA sequencing is PRJNA1135964.

The following datasets were generated:

| Author(s) | Year | Dataset title | Dataset URL | Database and Identifier |
|---|---|---|---|---|
| Lejeune A, Shopsin B, Cadwell K | 2024 | 16s rRNA mouse microbiome | https://www.ncbi.nlm.nih.gov/bioproject/PRJNA1135964 | NCBI BioProject, PRJNA1135964 |
| Lejeune A, Shopsin B, Cadwell K | 2024 | RNA seq of cecla tissue *Mus musculus* | https://ncbi.nlm.nih.gov/sra/PRJNA1134782 | NCBI Sequence Read Archive, PRJNA1134782 |

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
