## [Editor Report · eLife Assessment]

This **fundamental** study highlights potential mechanisms underlying the sex-dependent bias in susceptibility to gut colonization by Methicillin-resistant *Staphylococcus aureus* (MRSA). The evidence supporting the conclusion is **compelling**. The work will interest biologists who study intestinal infection and immunity.

---

## [Referee Report · Reviewer #1 (Public review)]

Summary:

Lejeune et al. demonstrated sex-dependent differences in the susceptibility to MRSA infection. The authors demonstrated the role of the microbiota and sex hormones as potential determinants of susceptibility. Moreover, the authors showed that Th17 cells and neutrophils contribute to the sex hormone-dependent protection in female mice.

Strengths:

The role of microbiota was examined in various models (germ-free, co-housing, microbiota transplantation). The identification of responsible immune cells was achieved using several genetic knockouts and cell-specific depletion models. The involvement of sex hormones was clarified using ovariectomy and the FCG model.

Weaknesses:

The specific microbial species/strains responsible for the protection, as well as the mechanisms by which these bacteria regulate sex hormone-mediated protection, remain unclear. However, this does not diminish the conceptual significance of the study.

Comments on revisions:

The authors have adequately addressed my previous concerns, and the revised manuscript shows significant improvement.

---

## [Referee Report · Reviewer #3 (Public review)]

Summary:

Using a mouse model of *Staphylococcus aureus* gut colonization Lejeune et al demonstrate that the microbiome, immune system, and sex are important contributing factors for whether this important human pathogen persists in the gut. The work begins by describing differential gut clearance of *S. aureus* in female B6 mice bred at NYU compared to those from Jackson Laboratories (JAX). NYU female mice cleared *S. aureus* from the gut but NYU male mice and mice of both sexes from JAX exhibited persistent gut colonization. Further experimentation demonstrated that differences between staphylococcal gut clearance in NYU and JAX female mice were attributed to the microbiome. However, NYU male and female mice harbor similar microbiomes, supporting the conclusion that the microbiome cannot account for the observed sex-dependent clearance of *S. aureus* gut colonization. To identify factors responsible for female clearance of *S. aureus*, the authors performed RNAseq on intestinal epithelia cells and cells enriched within the lamina propria. This analysis revealed sex-dependent transcriptional responses in both tissues. Genes associated with immune cell function and migration were distinctly expressed between the sexes. To determine which immune cell types contribute to *S. aureus* clearance Lejeune et al employed genetic and antibody-mediated immune cell depletion. This experiment demonstrated that CD4+ IL17+ cells and neutrophils promote elimination of *S. aureus* from the gut. Subsequent experiments, including the use of the 'four core genotype model' were conducted to discern between the roles of sex chromosomes and sex hormones. This work demonstrated that sex-chromosome linked genes are not responsible for clearance, increasing the likelihood that hormones play a dominant role in controlling *S. aureus* gut colonization.

Strengths:

A strength of the work is the rigorous experimental design. Appropriate controls were executed and, in most cases, multiple approaches were conducted to strengthen the authors' conclusions. The conclusions are supported by the data.

The following suggestions are offered to improve an already strong piece of scholarship.

Weaknesses:

The correlation between female sex hormones and elimination of *S. aureus* from the gut could be further validated by quantifying sex hormones produced in the four core genotype mice in response to colonization. Additionally, and this may not be feasible, but according to the proposed model administering female sex hormones to male mice should decrease colonization. Finally, knowing whether the quantity of IL-17a CD4+ cells change in the OVX mice has the potential to discern whether the abundance/migration of the cells or their activation is promoted by female sex hormones.

In the Discussion the authors highlight previous work establishing a link between immune cells and sex hormone receptors, but whether the estrogen (and progesterone) receptor is differentially expressed in response to *S. aureus* colonization could be assessed in the RNAseq dataset. Differential expression of known X and Y chromosome linked genes were discussed but specific sex hormones or sex hormone receptors, like the estrogen receptor were not. This potential result could be highlighted.

Comments on revisions:

The authors have adequately addressed my comments. I have only one minor adjustment: the Esr1 mice should be included the Materials and Methods.

---

## [Author Response]

The following is the authors’ response to the original reviews.

**Reviewer #1 (Public review):**
Summary:Lejeune et al. demonstrated sex-dependent differences in the susceptibility to MRSA infection. The authors demonstrated the role of the microbiota and sex hormones as potential determinants of susceptibility. Moreover, the authors showed that Th17 cells and neutrophils contribute to sex hormone-dependent protection in female mice.Strengths:The role of microbiota was examined in various models (gnotobiotic, co-housing, microbiota transplantation). The identification of responsible immune cells was achieved using several genetic knockouts and cell-specific depletion models. The involvement of sex hormones was clarified using ovariectomy and the FCG model.Weaknesses:The mechanisms by which specific microbiota confer female-specific protection remain unclear.

We thank the reviewer for highlighting the strengths of the manuscript including the models and techniques we employ. We agree that the relationship between the microbiota and sex-dependent protection is less developed compared with other aspects of the study. As detailed below, we are attempting to identify specific microbes that confer femalespecific protection and links with sex hormones. We have promising but preliminary results. Thus, in our revised manuscript, we added new data on the host response as suggested by the detailed comments from the Reviewers. We also elaborate on the potential role of the microbiota in the discussion section.

**Reviewer #1 (Recommendations for the authors):**
(1) The authors nicely showed that the transfer of the protective phenotype by FMT requires the female sex in recipients (Figure 2E). However, it remains unclear whether the female sex is required to develop protective microbiota in donor mice, as only the female NYU donor-male Jax recipient combination was tested. What happens if the microbiota from male NYU mice is transplanted into female Jax mice? If sex hormones act only on the downstream of the microbiota, such mice would show the protective phenotype. However, if sex hormones are required to establish a protective microbiota, the transplantation of microbiota from male NYU mice will not confer protection in recipient female Jax mice.

The Reviewer’s comment is well taken. We have not conducted the suggested experiment of FMT from male NYU mice to JAX female mice yet because we are pursuing an in vitro approach that we hope will eventually provide a more definitive answer. We observed that stool from female NYU mice and not JAX mice inhibits MRSA when cultured under anaerobic conditions, and this inhibitory activity is eliminated by filtration (Author response image 1A). We also observed that stool from male NYU mice inhibits MRSA growth to a similar extent as stool from female NYU mice (Author response image 1B). This result suggests that the protective role of sex hormones is downstream of the microbiota. We are in the process of identifying the specific microbiota member to support this conclusion.

**Author response image 1. sa3fig1:** Stool from NYU mice inhibits MRSA growth *in vitro*. (A) MRSA CFU/mL in media (TSB) following culture with unfiltered or filtered stool homogenate from female NYU or JAX mice. Stool homogenate or TSB alone was added in a 1:1 ratio to 1x106 CFU/mL MRSA and cultured anaerobically for up to 24 hours. (B) MRSA CFU/mL in TSB following culture with unfiltered stool homogenate from NYU male or female mice. Stool homogenate or TSB alone was added in a 1:1 ratio to 1x106 CFU/mL MRSA. 3 experimental replicates performed; stool taken from 6 individual mice per condition. Mean MRSA burden ± SEM. Area under the curve analysis + One way ANOVA with Sidak’s multiple comparisons test. ns: not significant.

(2) The results clearly showed the involvement of the specific microbiota in NYU mice in the sex-dependent bias in susceptibility to MRSA. However, the mechanisms by which specific microbiota promotes female sex-mediated protection need to be better described. Is this simply attributed to the different Th17 cell numbers in NYU and Jax mice (i.e., increased commensalspecific Th17 cells in NYU like Taconic mice)? Or is it possible that NYU microbiota impacts the regulation of sex hormones or their downstream signaling? What about the level of sex hormones in NYU and Jax mice? Are these levels equivalent or different? Do NYU and Jax microbiotas regulate the expression of sex hormone receptors in immune cells differently?

These are great questions. We do not observe baseline differences in Th17 cells like JAX versus Taconic mice (Figure 5B), suggesting that the mechanism is different. However, it is quite possible that an antigen-specific T cells, or Th17 cell specifically, is present at low levels and expands rapidly upon MRSA colonization. We have added this possibility to the discussion in the revised manuscript. To address the Reviewer’s question about the effect of the microbiota on sex hormones, we first sought to determine which sex hormone is necessary. Using estrogen receptor knockouts (*Esr1-/-*), we were able to implicate estrogen and have added this important finding to the manuscript (Fig 6C). Then, we measured levels of estradiol in stool samples but did not observe a difference between NYU and JAX female mice (Author response image 2). We provide the results below but did not add it to the revised manuscript because we found it difficult to draw a conclusion without more extensive profiling as well as quantification of the receptor on specific immune cell subsets and cell-type specific knockouts. Also, see our response to Reviewer #3 regarding receptor expression. Although we have yet to explain the role of the microbiota, we hope the Reviewer agrees that we have promising yet preliminary results and that the new experiments we added to the manuscript have further strengthened the mechanism on the host-side.

**Author response image 2. sa3fig2:** Estradiol levels in stool samples prior to MRSA inoculation. (A) Estradiol levels in stool samples collected prior to MRSA inoculation in male and female mice bred at NYU or purchased from Jackson Labs. Frozen stool samples were normalized by weight and processed using the DetectX Estradiol ELISA Kit (Arbor Assays).

(3) The authors claimed that Th17-mediated recruitment of neutrophils likely promotes the clearance of MRSA in female NYU mice. However, the experimental evidence supporting this claim could be stronger. The authors should show the neutrophil recruitment in the gut mucosa in female and male NYU mice. Also, the levels of neutrophils between NYU and Jax female mice should be examined. To further strengthen the link between Th17 and neutrophils, it would be ideal to analyze neutrophil recruitment in mice lacking Th17 cells (i.e., Rag2-/-, anti-CD4 treated, Rorgt-/- mice).

We agree and now include a more detailed analyses of neutrophils. We found that the number of neutrophils in the intestine were not higher in NYU female mice compared with NYU male mice, with or without MRSA. Instead, we show that neutrophils in NYU female mice display higher levels of surface CD11b, a sign of activation, compared to males following inoculation with MRSA . We have added these findings to the revised manuscript (Fig5 H and I). IL-17 can activate neutrophils and increase their antimicrobial activity. Consistent with this possibility, we now show that female mice lacking the IL-17 receptor lose the enhanced colonization resistance. Based on these findings, we have modified this aspect of the conclusion, and thank the reviewer for the helpful suggestion.

**Reviewer #2 (Public review):**
The current study by Lejeune et al. investigates factors that allow for persistent MRSA infection in the GI tract. They developed an intriguing model of intestinal MRSA infection that does not use the traditional antibiotic approach, thereby allowing for a more natural infection that includes the normal intestinal microbiota. This model is more akin to what might be expected to be observed in a healthy human host. They find that biological sex plays a clear role in bacterial persistence during infection but only in mice bred at an NYU Facility and not those acquired from Jackson Labs. This clearly indicates a role for the intestinal microbiome in affecting female bacterial persistence but not male persistence which was unaffected by the origin of the mice and thus the microbiome. Through a series of clever microbiome-specific transfer experiments, they determine that the NYU-specific microbiome plays a role in this sexual dimorphism but is not solely responsible. Additional experiments indicate that Th17 cells, estrogen, and neutrophils also participate in the resistance to persistent infection. Notably, they assess the role of sex chromosomes (X/Y) using the established four core genotype model and find that these chromosomes appear to play little role in bacterial persistence.Overall, the paper nicely adds to the growing body of literature investigating how biological sex impacts the immune system and the burden of infectious disease. The conclusions are mostly supported by the data although there are some aspects of the data that could be better addressed and clarified.

We thank the Reviewer for appreciating our contribution and these supportive comments. We have added several experiments to fill-in gaps and text revisions to increase clarity and acknowledge limitations.

(1) There is something of a disconnect between the initial microbiome data and the later data that analyzes sex hormones and chromosomes. While there are clearly differences in microbial species across the two sites (NYU and JAX) how these bacterial species might directly interact with immune cells to induce female-specific responses is left unexplored. At the very least it would help to try and link these two distinct pieces of data to try and inform the reader how the microbiome is regulating the sex-specific response. Indeed, the reader is left with no clear exploration of the microbiota's role in the persistence of the infection and thus is left wanting.

We agree. This comment is similar to Reviewer #1’s feedback. As mentioned above, we are attempting to clarify the association between sex differences and the microbiota and have included preliminary results for the Reviewers. However, addressing this disconnect will require substantially more investigation. Instead, we have added insightful new data that elaborate on aspects of the host response. We hope the Reviewer agrees that revised manuscript is stronger and that further delineation of the microbiota can be addressed by future studies.

(2) While the authors make a reasonable case that Th17 T cells are important for controlling infection (using RORgt knockout mice that cannot produce Th17 cells), it is not clear how these cells even arise during infection since the authors make most of the observations 2 days postinfection which is longer before a normal adaptive immune response would be expected to arise. The authors acknowledge this, but their explanation is incomplete. The increase in Th17 cells they observe is predicated on mitogenic stimulation, so they are not specific (at least in this study) for MRSA. It would be helpful to see a specific restimulation of these cells with MRSA antigens to determine if there are pre-existing, cross-reactive Th17 cells specific for MRSA and microbiota species which could then link these two as mentioned above.

We acknowledge that this is a limitation of our study. Although an experiment demonstrating pre-existing, cross-reactive T cells would help support our conclusion, aspects of MRSA biology may make the results of this experiment difficult to interpret. We have consulted with an expert on MRSA virulence factors, co-lead author Dr. Victor Torres, about the feasibility of this experiment. MRSA possess superantigens, such as Staphylococcal enterotoxin B, which bind directly to specific Vβ regions of T-cell receptors (TCR) and major histocompatibility complex (MHC) class II on antigen-presenting cells, resulting in hyperactivation of T lymphocytes and monocytes/macrophages. Additionally, other MRSA virulence factors, such as α-hemolysin and LukED, induce cell death of lymphocytes. MRSA’s enterotoxins are heat stable, so heat-inactivation of the bacterium may not help in this matter. For these reasons, it is unlikely that we can perform a simple restimulation of lymphocytes with MRSA antigens.

A study by Shao et al. provides an example of a host commensal species inducing Th17 cells with cross-reactivity against MRSA. Upon intestinal colonization, the intestinal fungus *Candida albicans* influences T cell polarization towards a Th17 phenotype in the spleen and peripheral lymph nodes which provided protection to the host against systemic candidemia. Interestingly, this induction of protective Th17 cells, increased IL-17 and responsiveness in circulating Ly6G+ neutrophils also protected mice from intravenous infection with MRSA, indicating that T cell activation and polarization by intestinal *C. albicans* leads to non-specific protective responses against extracellular pathogens.

Shao TY, Ang WXG, Jiang TT, Huang FS, Andersen H, Kinder JM, Pham G, Burg AR, Ruff B, Gonzalez T, Khurana Hershey GK, Haslam DB, Way SS. Commensal Candida albicans Positively Calibrates Systemic Th17 Immunological Responses. Cell Host & Microbe. 2019 Mar 13;25(3):404-417.e6. doi: 10.1016/j.chom.2019.02.004. PMID: 30870622; PMCID: PMC6419754.

We have added a brief version of the above discussion in the revised manuscript. Also, as mentioned earlier, we have added new data strengthening the axis between Th17 and neutrophils, including showing that IL-17 receptor is necessary and that neutrophils display signs of heightened activation in female mice during MRSA colonization.

(3) The ovariectomy experiment demonstrates a role for ovarian hormones; however, it lacks a control of adding back ovarian hormones (or at least estrogen) so it is not entirely obvious what is causing the persistence in this experiment. This is especially important considering the experiments demonstrating no role for sex chromosomes thus demonstrating that hormonal effects are highly important. Here it leaves the reader without a conclusive outcome as to the exact hormonal mechanism.

This is a great suggestion. Rather than adding back ovarian hormones, we performed the more direct experiment and tested whether the estrogen receptor (ERα, encoded by *Esr1*) is necessary for the enhanced colonization resistance. Indeed, we observed that *Esr1-/-* female mice have increased MRSA burden compared to *Esr1+/-* littermates. We have added this new result (Figure 6C) and thank the Reviewer for their guidance.

1. The discussion is underdeveloped and is mostly a rehash of the results. It would greatly enhance the manuscript if the authors would more carefully place the results in the context of the current state of the field including a more enhanced discussion of the role of estrogen, microbiome, and T cells and how the field might predict these all interact and how they might be interacting in the current study as well.

Author response: We thank the Reviewer for their feedback in improving the scholarship on the manuscript. We have expanded on the literature and the mechanistic model in both the discussion section and other parts to provide better context for our findings.

**Reviewer #3 (Public review):**
Summary:Using a mouse model of *Staphylococcus aureus* gut colonization, Lejeune et al. demonstrate that the microbiome, immune system, and sex are important contributing factors for whether this important human pathogen persists in the gut. The work begins by describing differential gut clearance of *S. aureus* in female B6 mice bred at NYU compared to those from Jackson Laboratories (JAX). NYU female mice cleared *S. aureus* from the gut but NYU male mice and mice of both sexes from JAX exhibited persistent gut colonization. Further experimentation demonstrated that differences between staphylococcal gut clearance in NYU and JAX female mice were attributed to the microbiome. However, NYU male and female mice harbor similar microbiomes, supporting the conclusion that the microbiome cannot account for the observed sex-dependent clearance of S. aureus gut colonization. To identify factors responsible for female clearance of *S. aureus*, the authors performed RNAseq on intestinal epithelial cells and cells enriched within the lamina propria. This analysis revealed sexdependent transcriptional responses in both tissues. Genes associated with immune cell function and migration were distinctly expressed between the sexes. To determine which immune cell types contribute to *S. aureus* clearance Lejeune et al employed genetic and antibody-mediated immune cell depletion. This experiment demonstrated that CD4+ IL17+ cells and neutrophils promote the elimination of *S. aureus* from the gut. Subsequent experiments, including the use of the 'four core genotype model' were conducted to discern between the roles of sex chromosomes and sex hormones. This work demonstrated that sex-chromosome-linked genes are not responsible for clearance, increasing the likelihood that hormones play a dominant role in controlling *S. aureus* gut colonization.Strengths:A strength of the work is the rigorous experimental design. Appropriate controls were executed and, in most cases, multiple approaches were conducted to strengthen the authors' conclusions. The conclusions are supported by the data.The following suggestions are offered to improve an already strong piece of scholarship.Weaknesses:The correlation between female sex hormones and the elimination of *S. aureus* from the gut could be further validated by quantifying sex hormones produced in the four core genotype mice in response to colonization. Additionally, and this may not be feasible, but according to the proposed model administering female sex hormones to male mice should decrease colonization. Finally, knowing whether the quantity of IL-17a CD4+ cells change in the OVX mice has the potential to discern whether abundance/migration of the cells or their activation is promoted by female sex hormones.In the Discussion, the authors highlight previous work establishing a link between immune cells and sex hormone receptors, but whether the estrogen (and progesterone) receptor is differentially expressed in response to *S. aureus* colonization could be assessed in the RNAseq dataset. Differential expression of known X and Y chromosome-linked genes were discussed but specific sex hormones or sex hormone receptors, like the estrogen receptor, were not. This potential result could be highlighted.

We appreciate the comment on the scholarship and thank the Reviewer for the insightful suggestions to improve this manuscript. We apologize for not including references that address some of the Reviewer’s questions. Other research groups have compared the levels of hormones between XX and XY males and females in the four core genotypes model and have found similar levels of circulating testosterone in adult XX and XY males. No difference was found in circulating estradiol levels in XX vs XY- females when tested at 4-6 or 79 months of age.

Karen M. Palaszynski, Deborah L. Smith, Shana Kamrava, Paul S. Burgoyne, Arthur P. Arnold, Rhonda R. Voskuhl, A Yin-Yang Effect between Sex Chromosome Complement and Sex Hormones on the Immune Response. Endocrinology, Volume 146, Issue 8, 1 August 2005, Pages 3280–3285, https://doi.org/10.1210/en.2005-0284

Sasidhar MV, Itoh N, Gold SM, Lawson GW, Voskuhl RR. The XX sex chromosome complement in mice is associated with increased spontaneous lupus compared with XY. Ann Rheum Dis. 2012 Aug;71(8):1418-22. doi: 10.1136/annrheumdis-2011-201246. Epub 2012 May 12. PMID: 22580585; PMCID: PMC4452281.

Administering female sex hormones to males is a good idea. We did not observe an effect of injecting males with estrogen on MRSA colonization (data not shown), perhaps due to the dose or timing, or because it is not sufficient (i.e., additional hormones and factors may be required). Therefore, we analyzed the necessity of estrogen signaling and found that *Esr1-/-* female mice impairs colonization resistance to MRSA. We have added this new experiment to the revised manuscript (Fig6 C).

Examination of the levels of estrogen, progesterone, and androgen receptors in our cecalcolonic lamina propria RNA-seq dataset is an excellent idea. We observed a significant increase in the G-protein coupled estrogen receptor 1 (*Gper1*) and a non-significant increase in Estrogen receptor alpha (Esr1) following MRSA inoculation in the immune cell compartment. This analysis has been added to the revised manuscript (Supplemental Fig6).

**Reviewer #3 (Recommendations for the authors)**
Minor editing issues:The topic sentence of the last paragraph in the Results section states - 'male sex defining gene sex determining region Y (Sry) has been moved from the Y chromosome to an autosome'. 'Sex defining gene' and sex-determining region seems redundant in this context. A sex-defining gene would presumably be located within a sex-determining region.Bold the letter 'F' in the Figure 5 legend.It's not clear from the Figure 6E legend when the IL-17A+ CD4+ cells were quantified, 2 dpi?In the third sentence of the second paragraph of the Discussion, the two references are merged together.

We thank the Reviewer for pointing out these editing issues. They have been addressed in the revised manuscript.